# The Epidemiology and Clinical Presentations of Atopic Diseases in Selective IgA Deficiency

**DOI:** 10.3390/jcm10173809

**Published:** 2021-08-25

**Authors:** Izabela Morawska, Sara Kurkowska, Dominika Bębnowska, Rafał Hrynkiewicz, Rafał Becht, Adam Michalski, Hanna Piwowarska-Bilska, Bożena Birkenfeld, Katarzyna Załuska-Ogryzek, Ewelina Grywalska, Jacek Roliński, Paulina Niedźwiedzka-Rystwej

**Affiliations:** 1Department of Clinical Immunology and Immunotherapy, Medical University of Lublin, Chodźki 4a St., 20-093 Lublin, Poland; izabelamorawska19@gmail.com (I.M.); adam.96.michalski@gmail.com (A.M.); ewelina.grywalska@gmail.com (E.G.); jacek.rolinski@gmail.com (J.R.); 2Department of Nuclear Medicine, Pomeranian Medical University, Unii Lubelskiej 1 St., 71-252 Szczecin, Poland; sarakurkowska95@gmail.com (S.K.); hannap@pum.edu.pl (H.P.-B.); birka@pum.edu.pl (B.B.); 3Institute of Biology, University of Szczecin, Felczaka 3c St., 71-412 Szczecin, Poland; dominika.bebnowska@usz.edu.pl (D.B.); rafal.hrynkiewicz@usz.edu.pl (R.H.); 4Clinical Department of Oncology, Chemotherapy and Cancer Immunotherapy, Pomeranian Medical University of Szczecin, Unii Lubelskiej 1, 71-252 Szczecin, Poland; rafal.becht@pum.edu.pl; 5Department of Pathophysiology, Medical University of Lublin, Jaczewskiego 8b St., 20-090 Lublin, Poland; zaluskakatarzynaa@gmail.com

**Keywords:** atopic diseases, atopy, allergy, selective IgA deficiency, primary immunodeficiency

## Abstract

Selective IgA deficiency (sIgAD) is the most common primary immunodeficiency disease (PID), with an estimated occurrence from about 1:3000 to even 1:150, depending on population. sIgAD is diagnosed in adults and children after the 4th year of age, with immunoglobulin A level below 0.07 g/L and normal levels of IgM and IgG. Usually, the disease remains undiagnosed throughout the patient’s life, due to its frequent asymptomatic course. If symptomatic, sIgAD is connected to more frequent viral and bacterial infections of upper respiratory, urinary, and gastrointestinal tracts, as well as autoimmune and allergic diseases. Interestingly, it may also be associated with other PIDs, such as IgG subclasses deficiency or specific antibodies deficiency. Rarely sIgAD can evolve to common variable immunodeficiency disease (CVID). It should also be remembered that IgA deficiency may occur in the course of other conditions or result from their treatment. It is hypothesized that allergic diseases (e.g., eczema, rhinitis, asthma) are more common in patients diagnosed with this particular PID. Selective IgA deficiency, although usually mildly symptomatic, can be difficult for clinicians. The aim of the study is to summarize the connection between selective IgA deficiency and atopic diseases.

## 1. Introduction

Primary immunodeficiency diseases (PIDs) are a heterogeneous group of congenital diseases with various clinical manifestations and different models of inheritance (X-linked, AR, polygenetic), caused by the impairment or loss of at least one function of the immune system. They weaken the body’s defenses, increasing the frequency of infections as well as the risk of autoimmune and proliferative diseases, including cancers [1]. 

PIDs can affect various elements of the immune system. As a result of next-generation sequencing and a better understanding of the molecular and immunological mechanisms, which affect the immune system, researchers can identify new genes and disorders. According to the latest data, ten basic types of PID can be distinguished: humoral and cellular response deficiency, PID’s with associated or syndromic features, predominantly antibody deficiencies, immune dysregulation, congenital defects of phagocyte number and/or function [2].

Early diagnosis is of major importance and might be life-saving in patients with some PID. Recurrent or severe infections should raise a suspicion for immunodeficiency. The National Primary Immunodeficiency Resource Center developed a list of ten warning signs of PID [3]. Besides, Cunningham-Rundles et al. developed an immunodeficiency-related (IDR) score to assess the likelihood of finding immunodeficiency [4]. According to the recent work of Bahrami et al. the mean diagnostic delay among primary immunodeficient patients was 2.05 ± 1.7 years [5]. This delay is especially prominent in antibody deficiency defects and therefore requires special attention. 

An unusual and challenging disease in the group of antibody deficiencies is selective IgA deficiency (sIgAD). Selective IgA deficiency is the most common primary immunodeficiency disease with an estimated occurrence from about 1:3000 to even 1:150, depending on the population, diagnosed more often in males [6,7]. The course of the disease is very varied, as most cases are asymptomatic, but recurrent infections, allergies, autoimmune diseases, and an increased risk of cancer may occur [7,8]. Besides the decreased level of serum IgA, patients with sIgAD suffer also from a deficiency of secretory IgA [9]. This facilitates the passage through the mucosal barrier for aeroallergens and food antigens, which makes these patients prone to develop allergies. Sometimes allergies can be even the first presentation of sIgAD. Aghamohammadi and colleagues reported that 40.5% of patients had allergic symptoms as the first manifestation of the disease [10]. Therefore, the suspicion of sIgAD should raise not only patients with recurrent infections but also with other clinical manifestations.

## 2. IgA—Structure

IgA is a class of immunoglobulins characterized by the presence of an alpha heavy chain. The daily synthesis of immunoglobulin A exceeds the total production of all other immunoglobulins [11]. In the human body, there are two subclasses of this immunoglobulin: IgA1 and IgA2. The most important difference between them lays in the structure of their hinge region and the number of the glycosylation sites [12]. 

In serum, IgA1 is predominant, accounting for as much as 90%, while in mucosal tissues, both subclasses are more evenly distributed, comprising 40% IgA1 and 60% IgA2. [13]. In human blood, IgA occurs mostly in monomeric form, while secretory IgA (SIgA) present on the surface of mucous membranes usually occurs in the form of dimers, much less often as trimers and tetramers [14,15]. Dimeric SIgA antibodies, covalently linked by a J-linking chain, are secreted onto the mucosal surface with their characteristic secretory complement (SC) [15].

## 3. IgA—Function

The majority of total IgA in the human body occurs in the mucosal tissues with a proven great role in the immune response. Serum level of IgA is 2–3 mg/mL, and it is the second most prevalent circulating immunoglobulin after IgG. However, until recently, the role of plasma IgA was still unclear. Now, we have some evidence that serum IgA has some immunological functions, which are independent of the role of secretory IgA. 

Serum monomeric immunoglobulin A acting through Fc alpha receptor I (FcαRI) has important immunomodulatory functions [16,17]. FcαRI is expressed on cells of the myeloid lineage, including monocytes, neutrophils, eosinophils, some macrophages, intestinal dendritic cells, and Kupffer cells [18]. Its role is associated with activation of different signaling pathways, immunoreceptor tyrosine-based activation motif (ITAM), and ITAM inhibitory (ITAMi) [19]. Once a multimeric ligand binds FcαRI, activation of an inflammatory response through ITAM signaling takes place. On contrary, monovalent ligand, like monomeric IgA, acts through ITAMi signaling, which results in an anti-inflammatory response (Figure 1).

Instead, secretory IgA due to such a numerous representation within the mucous membranes is considered the body’s first line of defense against harmful external factors [20]. It has been proven that they can eliminate pathogens, for example, by adsorption of food antigens, agglutination of bacteria, inhibition of epithelial adhesion to mucous membranes [16,18,21]. Reports indicate the ability to neutralize and inhibit the release of viral and bacterial particles, neutralize toxins and enzymes produced by numerous pathogens [22,23]. 

IgA has been shown to exert an anti-inflammatory function by inducing the expression of anti-inflammatory cytokines such as IL-10 and inhibiting releasing pro-inflammatory cytokines such as IL-6 [24,25,26]. Moreover, IgA activates complement only in a limited amount, but this class of immunoglobulin can block the activation of the complement mediated by IgG [21,27]. It is known that IgA silences some responses after bacterial cell activation, like for example, oxidative burst activity, phagocytosis, as well as chemotaxis [28]. Effector functions of IgA are complex. As mentioned before, IgA may interfere directly with immune cells of the myeloid lineage using FcαRI. After the interaction of IgA with monocyte-derived dendritic cells, antigen presentation, maturation, and production of IL-10 may occur [29]. Monocytes also have the ability to produce IL-10 after IgA ligation and to inhibit the production of IL-6 and TNF-alfa [30]. SIgA is important in eosinophil activation and degranulation and is more potent at stimulation of the release of reactive oxygen species than IgE, as well as it regulates oxidative burst and cytokine release by human alveolar macrophages [31,32]. Moreover, binding IgA with mannan-binding lectin (MBL) results in complement activation, which is a part of antimicrobial defense [33]. SIgA may act as a competitive blocker of IgG-mediated complement activation [30]. A significant aspect influencing the proper development of humoral immunity is an adequate stimulation of the immune system and, as a result, the ability to maintaining an appropriate balance between the cellular and humoral response.

## 4. Pathogenesis of IgA Deficiency

The pathogenesis of the disease is not yet fully understood. It is possible that sIgAD can be caused by the overlap of some of these mechanisms [34].

One of them, which has been widely described, is the presence of errors in the differentiation of IgA^+^ plasmablasts causing a low number of IgA-secretory cells, difficulties with switching IgA to SIgA, and a low number of mRNA in B-cells producing IgA [7,35,36,37,38]. Another mechanism concerns cytokines that are involved in IgA production (IL-10, IL-4, IL-6, IL-12, IL-21) and is caused by dysregulation of their pathways, especially in secondary lymphoid organs [8,9,39,40,41,42,43,44]. In patients with sIgAD, IL-10 is proven to be crucial in the differentiation of the B cells to IgA-secreting cells. Furthermore, it has a synergic effect with IL-4 [43,45]. Another cytokine that causes IgA production is TNF-B, it also possesses the ability to act as an isotype “switch” factor for IgA production [46,47]. Lowered level of TGF-B may lead to a low IgA level in patients with sIgAD [9]. Il-21 stimulation is even more effective in inducing IgA production than IL-4 and IL-10, as well as it prevents CD19 + B cells spontaneous apoptosis [39]. This increased apoptosis could cause a reduction in survival of B cells and, therefore, decreased production of normal levels of IgA immunoglobulin [48]. Additionally, it is possible that T-cells impairment is connected with sIgAD. Soheili et al. suggest a direct association between decreased level of T regulatory (T_reg_) and the severity of clinical presentation of sIgAD [49]. In this study, patients were divided into two groups—G1 with a lower-than-cut-point T_reg_ value, where there was a higher risk of developing autoimmunity and class switching recombination defects, and G2 with a higher T_reg_ value, where only one person had autoimmunity and no one had the described antibodies defect. The link between T_reg_ cells and IgA production is complex and multifaceted. T_reg_ cells colonize the intestinal mucosa where they produce TGF-beta and IL-10, which are essential in the production of IgA. Reduced amount of T_reg_ negatively affects the amount of IgA^+^ B lymphocytes, and restoration of the correct amount of T_reg_, consequently, restores normal IgA production in the intestines [50,51,52] Interestingly, according to the meta-analysis by Bronson et al. there is a multiple gene linkage between the “Intestinal Immune Network for IgA production” and “T_reg_” [53]. Besides, the highest levels of APRIL (a proliferation-inducing ligand), which is connected with IgA-synthesis as a compensatory mechanism, are observed in patients with sIgAD [20,54]. It has been proved that there is a genetic background to sIgAD [55,56,57]. 

Moreover, level of these immunoglobulins may be influenced by drugs that are often used in everyday practice—non-steroid anti-inflammatory drugs (NSAIDs), angiotensin convertase enzyme inhibitors (ACEI), several anti-epileptic drugs, or drugs used in rheumatology. They can even trigger iatrogenic isolated sIgAD [58,59]. Moreover, some of the viral infections, e.g., EBV, hepatitis type C may induce post-infection IgA deficiency [60,61].

## 5. Clinical Presentation of sIgAD

Based on clinical presentation, sIgAD patients can be classified into different phenotypes. Yazdani et al. [8] in their work from 2015 divided these phenotypes into five main categories: asymptomatic, minor infections, autoimmunity, allergy, severe. It was reported that there is no correlation between serum IgA levels and clinical phenotype and disease severity [34]. 

Diagnosis of sIgAD is a diagnosis of exclusion. Immunologists should take into consideration infection-induced or drug-induced IgAD, as well as drug-induced IgAD/IgG2 subclass deficiency [62,63,64]. Important factors in establishing the diagnosis of IgA deficiency are family background and other laboratory parameters, which are relevant in order to differentiate sIgAD from CVID (lowered IgA/IgG and sometimes IgM levels), secondary hypogammaglobulinemia (moderately low levels of IgA), single-gene primary immunodeficiencies, hypoglobulinemia due to the protein loss as the result of enteropathy or nephrotic syndrome and malignancies such as thymoma, myeloma, and chronic lymphocytic leukemia [64,65].

### 5.1. Asymptomatic

Most of the patients are asymptomatic. The estimate number is 60% [8], but it varies in different studies. These patients might develop some clinical manifestations; therefore, they should undergo regular evaluations [34]. A 22-year study based on a cohort of 184 pediatric SIgAD patients, performed by Lougaris et al. [66] shows how the clinical presentation of the disease can vary with time. They assessed laboratory parameters and long-term health status of patients, 62% of whom had symptoms at the time of diagnosis. Allergic complications during follow-up were additionally developed in 16%, autoimmune diseases excluding celiac disease in 9%, and celiac disease developed in 11% of previously disease-free patients. During the follow-up period, 4% of patients achieved age-appropriate IgA levels, 9% of patients achieved partial IgA deficiency diagnosis and 2% of patients developed CVID.

### 5.2. Minor Infections

Children with recurrent and severe infections present a diagnostic challenge [67]. Four or more ear infections and two or more serious sinus infections or episodes of pneumonia within one year are warning signs for primary immunodeficiencies in children [68].

Secretory IgA plays an important role in maintaining the equilibrium of the body, as it takes part in the mucosal immune system and serves as the interface between the body and the microbiome. In the human body, the largest mucosal systems are the gastrointestinal tract and respiratory system, and therefore, decreased level of IgA will affect mostly them.

Symptomatic patients with minor infection usually present recurrent upper respiratory tract infections (40–90%), mainly viral, less frequently bacterial (with encapsulated bacteria etiology, such as Streptococcus pneumoniae, Hemophilus Influenzae) [7,8,69,70]. Bacterial bronchitis and pneumonia are much less common, but these infections may be complicated by bronchiectasis [71]. Infections of the ear, sinuses, conjunctiva, nose, and throat mucosa may occur. Most often, these infections are mild, not requiring hospitalization and their treatment does not differ significantly from the treatment of a patient without sIgAD. It was found that patients with sIgAD have a compensatory increase in secretory IgM level [9,72], however, these IgM cannot replace all functions of IgA [73]. In the intestines of sIgAD patients, there is 65–75% of Ig-containing plasma cells with the ability to produce IgM in comparison to about 6% in healthy volunteers, possibly due to the homology in structure and function between those two isotypes [74,75,76,77]. Besides, patients with sIgAD suffer from urinary tract infections (UTI) and gastrointestinal tract infections with viruses and bacteria. Moreover, intermittent or chronic diarrhea due to Giardia Lamblia is common, because the attachment and proliferation of this parasite on the gastrointestinal mucous are facilitated due to lack of IgA [78,79]. It is important to mention that in patients with recurrent UTI, bronchitis and pneumonia, defects in the urinary and respiratory systems should be excluded.

The diagnosis of selective IgA deficiency mostly does not significantly influence the therapeutic management of patients. Treatment of infections should be adequate to their etiology, patient’s age, and clinical condition. Treatment of comorbidities and prevention of complications remain the basis. There are no clear guidelines that would suggest the need for longer and more aggressive antibiotic therapy in this disease than in patients without sIgAD. There is also no consensus on the use of antibacterial prophylaxis in this immunodeficiency, but its usage was suggested in more severe cases, at least periodically [34,48,80]. Vaccinations play a significant role in minimizing the risk of infections [48]. It is advisable to extend the standard calendar with vaccination against *S. pneumoniae*, *N. meningitidis*, *H. influenzae,* and annual vaccination against influenza [34]. It is not typically recommended to initiate IgG (i.v, s.c) replacement therapy in patients without the coexistence of other immune-related diseases, acute, severe infections, or coexistence of specific antibodies deficiency [81]. According to one of the latest meta-analyses covering the effects of oral probiotics, parabiotics, and synbiotics on immunoglobulin levels, it has been shown that their supplementation increases significantly salivary IgA secretion, without a significant effect on the level of other immunoglobulins and with no effect on the serum IgA [82]. In addition, there are reports of an increase in the amount of IgA^+^ cells in the intestines of lamina propria in mice after oral ingestion of Lactobacillus-based preparations [83,84]. One prospective, randomized study demonstrated the validity of the use of oral immunomodulator bacterial extract (OM-85 BV) in patients with sIgAD and/or IgG subclass deficiency, resulting in a lower one-year infection rate [85]. A suggestion has been made to use oral IgA in patients with sIgAD, since this deficiency is associated with dysbiosis and chronic inflammation, and the present inflammation is inversely correlated with systemic anticommensal IgG response, which acts as “second line of defense” [86,87].

The importance of the IgA was raised again because of the ongoing global pandemic of coronavirus disease 2019 (COVID-19). When looking for the reasons for the varied course of the disease, questions arose on whether deficiency of IgA could be the reason for disease severity, vaccine failure, and prolonged viral shedding [88]. As mentioned above, the prevalence of sIgAD differs in various countries and the same was found for COVID-19. Naito et al. compared the number of cases of COVID-19 with the prevalence of selective IgA deficiency in different countries [89]. They found “a strong positive correlation between the frequency of sIgAD and the COVID-19 infection rate per population”. It was then concluded that one of the factors contributing to the low death rate from COVID-19 infection in Japan could be the low incidence of sIgAD in the country. As primary immunodeficiencies are a group of rare diseases, there is little data on the coexistence of sIgAD and COVID-19 infection. Nevertheless, literature data showing an extremely significant effect of class A immunoglobulins on early protection against SARS-CoV-2 virus also suggest a potentially more severe/complicated course of the disease [88,90]. This thesis is supported by the aforementioned literature data: a positive correlation between a high number of COVID-19 infections and a high incidence of sIgAD has been demonstrated, and an inverse relationship was observed in the extreme example of Japan [88]. In Israel, during two “so-called” waves, 20 patients with PID were affected by COVID-19 and none of them was diagnosed with sIgAD [91]; importantly, the relationship between the development of autoimmune diseases in the course of COVID-19 in patients with sIgAD—AIHA and Guillain-Baree syndrome [92]. Researchers also point to the risk of a poor response against SARS-CoV-2 after immunization in this group of patients [88].

### 5.3. Autoimmunity

There is an association between IgA deficiency and a higher prevalence of autoimmune disease [93,94]. Based on extended research in that field, the prevalence of autoimmune disease in this group rises to 31.7% [95]. According to Azizi et al. the median age of the onset of the first episode of autoimmunity was 7 [95]. Among diseases with higher prevalence in sIgAD subjects, we differentiate systemic lupus erythematosus, hypo- and hyperthyroidism, type 1 diabetes mellitus, Crohn’s disease, ulcerative colitis, rheumatoid arthritis, juvenile idiopathic arthritis, ankylosing spondylitis, and vitiligo. Whereas other diseases like scleroderma, celiac disease, autoimmune hepatitis, immune thrombocytopenic purpura, and autoimmune hemolytic anemia, occur less often but still with higher prevalence than in the general population [96].

The mechanism of autoimmunity in sIgAD is still not fully understood. There are six hypotheses that try to explain these phenomena, each based on a different mode of autoimmunity, such as human leukocyte antigen, cytogenic, monogenic, molecular mimicry, lingering inflammation and immune complexes, dysregulation of molecular pathways [96]. Some studies suggest that various mechanisms are likely to play concurrently. It has been found that there is also a higher incidence of autoimmunity in first-degree relatives of sIgAD patients [97].

In a recent case reported by Pfeuffer et al., authors stated that the presence of other acute diseases could induce autoimmunity in SIgAD patients [98]. In their case, it was Guillain-Barré syndrome induced by COVID-19.

### 5.4. Allergy

Allergy has long been a component of immune deficiency; however, allergic burden differs in different types of immunological disorders. Both PID and allergy are associated with impaired reactions of the immune system. In the case of PID, malfunctioning of some of its components will lead to infectious susceptibility. Atopy, on the other hand, is a hypersensitivity reaction of the immune system and a form of misdirected immunity.

The true prevalence of allergy among patients with sIgAD is still under debate since studies from different countries present inconsistent results. Therefore, it suggests that the prevalence varies depending on the ethnic background. There is even controversy in the scientific world whether the coexistence of IgA deficiency and allergic diseases is, in fact, true. Most publications support the relationship between sIgAD or low-IgA levels and allergic diseases [10,99,100,101,102,103], but some researchers deny it [104,105]. This connection has been the subject of medical research for over 50 years. In 1975, Buckley suggested that about half of the patients with sIgAD presented atopic diseases and related findings have been published later by Kemola [106,107]. A similar observation has been done in Ankara more recently, in 2017, where 45.7% of the patients diagnosed with sIgAD presented one of the following: asthma, rhinitis, eczema, atopic dermatitis, and interestingly the prevalence of allergy in a close family of this patients rose up to 43.2% [101]. Aghamohammadi, in his study on Iranian patients, revealed that allergy was observed in 84% of patients with sIgAD [10]. In a study from China, 17.6% of patients had allergic symptoms, however, most of them were allergic reactions to drugs (mostly penicillin) [108]. These results were inconsistent with typical allergies reported in other countries, such as asthma, rhinitis, food allergy, and atopic dermatitis [108]. On the other hand, there is a study with a prevalence of allergy in children with sIgAD on the percentage of 13% [70].

Many clinicians point out the frequent coexistence of IgA deficiency with bronchial asthma, allergic rhinitis, and atopic dermatitis in everyday practice but it is unclear whether it is the immunoglobulin A deficiency that promotes an allergic reaction, or the allergic reaction weakening the mucous membranes and consequently leading to a secondary IgA deficiency. There are plenty of possible explanations of this phenomenon. The connection between IgA deficiency and allergies may be caused by increased levels of circulating antigens, due to increased permeability at mucosal surfaces. It could also be a result of the inability to induce ITAMi signaling, due to decreased level of monomeric serum IgA, which, consequently, causes overactivation of the immunological system [79,109]. Another hypothesized mechanism is the deficiency of TGF-beta response. TGF-beta has properties to induce IgA synthesis, as well as inhibiting proliferation of Th2-cells. Th2-response is involved in the pathogenesis of atopic diseases [110,111,112]. Interestingly, there are allergen-specific A immunoglobulins, but their role in the pathogenesis of allergic diseases is unclear [113]. We do not know whether they are responsible for exacerbation or silencing the symptoms, but what we know is that they are observed in healthy people without allergic symptoms and low or undetectable IgE-levels [114,115]. Moreover, children with a tendency to allergic diseases have a more pronounced physiological IgA deficiency in the neonatal period and the lower these concentrations are, the greater is the severity of symptoms (although they usually remain within the reference values for age).

The American Academy of Allergy, Asthma & Immunology (AAAAI) and the American College of Allergy, Asthma & Immunology (ACAAI) developed practice parameters to guide the management of primary immunodeficiencies [116]. It is stated there that atopic diseases should be treated aggressively in patients with sIgAD. Since allergic inflammation facilitates the development of respiratory tract infections, it is crucial to treat allergy using all standard modalities, like avoidance of allergens, medication, and immunotherapy [116]. Our clinical experience shows that treating atopic diseases in patients with immunodeficiency is difficult and requires special attention and scrupulousness. In addition to the commonly used anti-histamine drugs, beta-mimetics, and glucocorticosteroids, in the case of treatment-resistant atopic diseases, biological drugs such as omalizumab and dupilumab might be helpful. Omalizumab is an anti-IgE antibody that is FDA approved for the treatment of moderate to severe allergic asthma, while dupilumab is an IL-4 receptor blocking antibody and is FDA approved for the treatment of moderate to severe atopic dermatitis in patients with the refractory disease [117,118,119]. The use of omalizumab in a young adult patient diagnosed with CVID, who suffered from chronic spontaneous urticaria, and did not respond either to an immunoglobulin substitution in immunomodulatory doses, anti-H1 and anti-H2 antihistamines, as well as leukotriene receptor antagonists, has been described. Only the inclusion of omalizumab resulted in a significant improvement in the condition of the skin and quality of life [120]. However, data suggest that such treatment could carry a risk of possible side effects—Banh et al. described a case of a 24-year-old patient diagnosed with asthma and CVID, where treatment with omalizumab might have increased the level of white blood cells and elevated myeloid cell count. Serious disorders, e.g., malignancy or severe infections were excluded. Importantly blood test results returned to normal levels shortly after drug discontinuation [121]. The use of dupilumab has been described in the context of a patient with CVID suffering from severe skin lesions such as erythematous-squamous and generalized infiltrated rash with exacerbation in sun-exposed zones and severe recurrent infections, in whom no improvement in skin condition was observed after treatment with glucocorticoids or cyclosporine. The introduction of dupilumab resulted in a reduction in the severity of skin lesions and the addition of IgG replacement therapy lowered the frequency of infections [122]. In our opinion, based on experience from other immunodeficiencies with predominantly antibody deficiency, it is possible to use the above-mentioned monoclonal antibodies in the treatment of severe allergic complications in patients with SIgAD.

#### 5.4.1. Food Allergy

The prevalence of food allergy in patients with PIDs was examined using the US Immunodeficiency Network (USIDNET). Surprisingly, it was lower than that in the general population. However, for some specific types of PID, like sIgAD, the prevalence was increased and it was found to be 25% [123], but there were only four patients with sIgAD in the registry. It is consistent with the study performed by Aghamohammadi et al. where the prevalence of food allergy among patients with sIgAD was 22% [10]. Across all studies in this review, the prevalence of food allergy among patients with SIgAD is presented in Table 1. Another study reports an increased risk of parentally reported food hypersensitivity at 4 years of age among children with sIgAD [124]. Moreover, the authors did not find any association between IgAD and increased levels of specific IgE, which could suggest that hypersensitivity in IgAD children is not IgE-mediated [124].

The majority of patients with deficiency of secretory IgA have substitution with secretory IgM. However, it might not guarantee proper mucosal protection and might allow food antigens to pass through the gastrointestinal mucosa and predispose to develop a food allergy. Another possible explanation connected with eczema and food allergy is the hypothesis that, due to the IgA-deficiency to gastrointestinal antigens in the gut, there is no antigen immunological-exclusion, which consists of antigen binding to SIgA at the level of the mucosal surface, and, consequently, blocking the absorption of the antigen [9,125].

Recent years showed that there is a strong connection between microbiota and allergy development. For example, in 2009, researchers found that children with allergy not only had lower salivary SIgA levels but also less differentiated bacterial microenvironment [126,127]. A study from 2018 focused on the effects of IgA deficiency on human gut microbiota composition [128]. They found out that patients with sIgAd have an altered gut microbiota composition compared to healthy patients. Moreover, the secretion of IgM cannot fully compensate for the lack of SIgA. It is therefore suggested that IgA plays a critical role in controlling stable gut microbial community. A different study from the same year showed only mild loss in microbial diversity in sIgAD subjects [129].

It was also found that serum IgA plays a role in suppressing IgE-mediated food allergy. IgE-mediated food allergy is a common cause of enteric disease, and, in the study conducted by Strait et al. concerning IgE-mediated systemic anaphylaxis induced by ingested allergens, it has been found that both serum antigen-specific IgG and IgA antibodies can protect against severe IgE-mediated allergic reaction [130]. This suggests that decreased serum IgA antibody levels might predispose to increased intestinal mucosal permeability and absorption of ingested antigens, therefore, increasing the risk of severe food allergy [131].

**Table 1 jcm-10-03809-t001:** Food allergy and sIgAD.

Year	Country	Sample Size	Disease Prevalence among SIgAD (%)	Reference	Diagnostic Tools
2009	Iran	37	22	[10]	The allergy status was evaluated by skin prick test, using 14 common standard allergen extracts
2012	Spain	330	4.2	[132]	Retrospective study of patients records
2017	Turkey	81	1.2	[101]	Skin prick tests + serum IgE measurements. Food allergy diagnosis was confirmed with an oral food challenge test.
2020	Iran	166	3.6	[133]	Data about clinical presentations were collected based on a detailed questionnaire

#### 5.4.2. Asthma

Asthma is a chronic inflammatory disease of the respiratory system characterized by bronchial hyperresponsiveness and reversible airflow obstruction. It is one of the most common chronic illnesses in childhood and its etiology in this group is vastly associated with atopy. Some studies report that asthmatic patients are more likely to have a diagnosis of sIgAD/CVID than non-asthmatic individuals [134]. In a study on an Iranian group, the prevalence of asthma among sIgAD patients was 51% [10], while in the general Iranian population it is 22–23% [135,136]. In the study on a Spanish group, asthma was observed in only 12.4% patients [97]. On the other hand, no difference in prevalence was found comparing sIgAD patients and control group in the case-control study of Jorgensen et al. [137]. The prevalence of asthma among patients with SIgAD is presented in Table 2.

Papadopoulou et al. state that the insufficient protection provided by the respiratory mucosa deprived of IgA in children with sIgAD makes them prone to develop bronchial hyperresponsiveness and consequently asthma [138]. In a different study, a high number of IgA-specific salivatory antibodies has been connected to a lower risk of late-onset wheezing in sensitized infants [139]. Furthermore, sIgAD may be connected with TNFRSF13B gene variants as one of the genetic susceptibilities. This gene encodes the transmembrane activator and calcium modulator and cyclophilin ligand interactor (TACI), which is the tumor necrosis factor receptor (TNFR) expressed on activated B cells and macrophages and is involved in isotype class switching to IgA [54,140,141]. Moschese et al. investigated the prevalence of TNFRSF13B mutations in 56 patients with absolute and partial sIgAD reporting 20% prevalence in this group [142]. Furthermore, researchers suggest that the mutation in these genes increases the risk of asthma development up to 2.5 fold, despite the IgE levels [69]. Moreover, the studies on the mice model proved that treating with antigen-specific IgA may protect animals from hyperresponsiveness as well as eosinophilic inflammation in airways [143]. Additionally, since mice do not express FcαRI [144], studies on human FcαRI transgenic mice were used in studies on the asthma model. It was found that by targeting FcαRI, IgA has been established as a strong inhibitor of asthma development [145].

Some studies reported a higher prevalence of respiratory tract infections among patients with sIgAD and allergy compared to those with sIgAD without any manifestation of allergy disease [99,133,146]. It suggests that allergic patients are more susceptible to respiratory tract infections.

**Table 2 jcm-10-03809-t002:** Asthma and sIgAD.

Year	Country	Sample Size	Disease Prevalence among SIgAD (%)	Reference	Diagnostic Tools
2008	Brazil	126	48.4	[79]	Diagnostic criteria of allergic diseases were not defined in the paper
2009	Iran	37	51	[10]	Lung function was evaluated according to the American Thoracic Society guidelines, using a computerized pneumotachograph
2010	Israel	63	23.8	[99]	Retrospective study of patients records
2012	Spain	330	12.4	[132]	Retrospective study of patients records
2013	USA	39	23	[134]	Asthma status was determined based on predetermined criteria for asthma
2013	Iceland	32	18.8	[137]	Self-administered questionnaire + interview performed by physician + lung function tests using spirometry
2017	Turkey	81	34.6	[101]	Asthma status was determined based on the Global Initiative for Asthma guidelines
2019	Italy	103	10.7	[142]	Patients’ clinical data were collected at enrolment and every 6–12 months for 5 years. Diagnostic criteria for allergic diseases were not defined in the paper
2020	Iran	166	6.6	[133]	Data about clinical presentations were collected based on a detailed questionnaire

#### 5.4.3. Atopic Dermatitis

A variety of primary immunodeficiencies have cutaneous manifestations. In the case of sIgAD, nonspecific cutaneous finding is eczematous dermatitis. Here, similarly to other allergic manifestations, there is a huge variety in the prevalence reported in different studies, which could be caused by ethnic diversity and, also, by different algorithms for atopic dermatitis diagnosis. Therefore, in a study performed by Aghamohammadi, the prevalence was 52% [10], but in the study of Magen, it was only 4.6%, however, it was still higher than in the control group [147]. The prevalence of atopic dermatitis among patients with SIgAD is presented in Table 3.

Moreover, Orivari et al., showed that the levels of secretory IgA in breast milk were inversely associated with the development of atopic dermatitis up to 2 and 4 years [148] among breastfeeding children. In a different study, though, such connection was not found [149].

Moreover, people with higher IgA levels and Staphylococcus aureus colonization in the gastrointestinal tract are less susceptible to the development of eczema [150].

**Table 3 jcm-10-03809-t003:** Atopic dermatitis and sIgAD.

Year	Country	Sample Size	Disease Prevalence among SIgAD (%)	Reference	Diagnostic Tools
2008	Brazil	126	2.4	[79]	Diagnostic criteria of allergic diseases were not defined in the paper
2009	Iran	37	49	[10]	The allergy status was evaluated by skin prick test, using 14 common standard allergen extracts
2010	Israel	63	3.2	[99]	Retrospective study of patients records
2012	Spain	330	3.6	[132]	Retrospective study of patients records
2015	Italy	102	57.84	[151]	Diagnosis was based on Hanifin-Rajka criteria and on skin biopsies where applicable
2017	Turkey	81	11.1	[101]	Diagnosis was based on Hanifin-Rajka criteria
2017	Israel	347	4.6	[147]	Retrospective study of patients records. Criteria for diagnosis of chronic spontaneous urticaria according to EAACI, GA2LEN, EDF and WAO guidelines
2019	Italy	103	12.6	[142]	Patients’ clinical data were collected at enrolment and every 6–12 months for 5 years. Diagnostic criteria for allergic diseases were not defined in the paper

#### 5.4.4. Allergic Rhinitis and Conjunctivitis

Serum IgA level in children under the age of 4 with positive skin-prick test was significantly lower than in healthy population, also allergic rhinitis and eczema were connected with a low level of salivary IgA [152]. The frequency of allergic rhinitis among patients with sIgAD in a study performed in Turkey was 27.2% [101], while the prevalence of allergic rhinitis in Turkish school-age children was 16.9% [153]. The presence of allergic rhinitis was only accepted if it was diagnosed by a physician. Furthermore, in a different study, the prevalence of allergic-rhinoconjunctivitis tended to be increased in the sIgAD group and was reported to be 37.5% [137]. Across all studies in this review, the prevalence of allergic rhinitis and conjunctivitis among patients with SIgAD is presented in Table 4.

### 5.5. Severe

As opposed to other primary immunodeficiencies, sIgAD rarely presents with severe manifestations. Therefore, differential diagnosis with other possible immunological disorders should be performed.

Patients with this phenotype suffer from recurrent and severe infections even in lower respiratory tracts [8]. One of the severe complications of severe respiratory infections is bronchiectasis. In such cases, it is crucial to eliminate other immunodeficiencies such as IgG2subclass, specific antibody deficiencies, and mannan-binding lectin deficiency [79].

Patients with this phenotype should be provided with extra care. In case of recurrent infections, prophylactic antibiotics should be considered, especially during autumn and winter [8]. The usage of IVIG replacement therapy in these patients is extremely controversial [116,154]. Usually, this treatment is recommended for individuals with both IgA deficiency and concomitant IgG2 subclass deficiency [155,156]. To determine if this treatment would be beneficial, the IgG antibody responses to protein and polysaccharide vaccines should be evaluated first [116].

## 6. Complications

In a prospective cohort study that examined mortality among patients with sIgAD turn out that they have an increased risk of death in the first 10 years after diagnosis [157]. Afterward, the mortality is similar to that of the general population. The most common causes of death include malignancy and cardiovascular diseases. There are a few life-threatening complications of sIgAD. Even if their prevalence is not high, they should be known for physicians to provide proper help for their patients. Among those included in the literature, we differentiate progression to CVID, transfusion-related anaphylaxis, and malignancy.

### 6.1. IgA Deficiency and CVID

Common variable immunodeficiency (CVID) is an immune disorder characterized by decreased serum levels of both IgG and IgA, with or without a decreased level of IgM, and poor antibody vaccine response or low switched memory B cells less than 70% of age-appropriate normal [158]. CVID most often presents with recurrent infections of the respiratory and gastrointestinal tract [159]. Symptomatic sIgAD and CVID have many similar features, moreover, some patients with sIgAD progress to CVID, especially if autoimmunity or IgG subclass deficiency is observed [149,150,151,152,153,154,155,156,157,158,159,160,161,162,163,164].

In patients with sIgAD, there is a significantly lower number of class-switched memory B cells and transitional B cells [165]. Preprint of another study showed an increased percentage of naive B cells and decreased percentage of switched memory B cells. Only one parameter correlated with the severity of the disease—CD21low cells. They were increased in patients with severe SIgAD as compared to those with mild severity [166]. Increased level of CD21low was previously described but without correlation to clinical status [167].

In CVID, there is a classification based on B-cell phenotype, which divides CVID patients into B − group and B + group, depending on the CD19 expression (lower or higher > 1%). B + patients may be further divided into groups smB + or smB-, based on a proportion of switched memory B-cell percentage (lower or higher >2%). Recently, an increase in transitional B cells and CD21low B-cells is used as a base to subdivide groups [168]. Some of the B-cell phenotype findings are similar between CVID and SIgAD. There is an interesting observation of an increased CD21low cells number in patients with severe sIgAD; a higher level of those cells in CVID patients is connected with autoimmune phenomena [169,170]. 

The major histocompatibility complex (MHC) represents the most common genetic susceptibility locus for CVID. However, non-MHC-associated single-gene mutations have been identified. These include the genes for ICOS, BAFF-R, TACI, CD19, CD21, CD81, CD20, LRBA, PKC-Delta, NF-kB1, NF-kB2, IL-21 [171]. Defects of these genes represent only approximately 2–10% of patients with CVID [172]. Some authors state that a common genetic basis for IgAD and CVID can suggest that at least in some cases, IgAD and CVID may be part of a spectrum of diseases caused by a common genetic factor—for example, a mutation in the TACI—transmembrane activator CAML (calcium modulator and cyclophilin ligand) [173]. Another, slightly different thesis is the presence of autoantibodies against BAFF, APRIL, or IL21 as a common ground for CVID and sIgAD [174]. Both in CVID and sIgAD, there are reports of an increased level of BAFF and APRIL [175]. Increased apoptosis is also one of the mechanisms reported for both of these disorders [48]. An interesting observation indicating a similar genetic background of both diseases is their coexistence in families [176].

Besides, the relation between human leukocyte antigen (HLA) A1, B8, DR3, DQ2, or any part of this haplotype and IgA deficiency could indicate progression to CVID [159,161]. In patients with severe clinical manifestations, HLA typing could be helpful for the prediction of progression to CVID [160]. Moreover, in sIgAD subjects with simultaneous Ig subclass deficiency and bronchiectasis, the presence of hematologic autoimmunity could be another predictor of progression to CVID [94].

### 6.2. Transfusion Selective IgA Deficiency

One of the most dangerous complications of sIgAD is an anaphylactic transfusion reaction. It has been found that some patients with sIgAD are sensitized, which means that there are anti-IgA antibodies (IgG or IgE) present in their blood [34,177,178,179]. These auto-reactive antibodies were found in 20–40% of patients with sIgAD [180]. After transfusion of blood containing IgA in such individuals, there is a risk of anaphylactoid reaction mediated by these immunoglobulins.

Rachid and Bonilla reviewed the articles reporting reactions to immunoglobulin products in patients with sIgAD [181]. The severity of adverse reactions with anti-IgA antibodies depends among others on the isotype (IgG or IgE), its specificity and serum concentration, the method of measurement. IgG anti-IgA antibodies are found in approximately one-third of sIgAD patients [181]. But only a few studies have reported anaphylactic transfusion reaction associated with IgE class. It has been also reported that IgE anti-IgA is less frequently studied than IgG anti-IgA. However, when both have been studied together, anti-IgA of the IgE class occurs much less frequently than IgG [182]. Burks et al. reported two patients (one with CVID and another with sIgAD) with IgE anti-IgA and IgG anti-IgA [177]. One of them had anaphylaxis with IVIG and another with IgA-deficient plasma. Ferreira et al. found IgE anti-IgA1 in a patient with CVID, which also had IgG anti-IgA [183].

The diagnosis of IgA-related anaphylaxis is made after transfusion-related anaphylaxis by measuring the levels of IgA and anti-IgA. The mechanism of anti-IgA production remains unexplained [184] and the clinically significant threshold of anti-IgA is still unknown [185]. However, the incidence of anaphylactic blood transfusion reactions is not very high and it occurs in one in 20,000–50,000 transfusions [186]. Moreover, some studies suggest that transfusion reactions occur less commonly than previously thought [187]. Only 17.5% of all blood samples coming from patients after transfusion reactions contained an IgA antibody, which indicates the presence of some other triggers [188].

There are no evidence-based guidelines regarding the proper approach while performing transfusion to patients with sIgAD. It is evident that patients with a history of anaphylactic transfusion reactions should not receive IgA-containing blood products [96]. These patients can receive blood products coming from donors with IgA deficiency, washed red blood cells, or platelet components. It is performed to remove residual plasma before transfusion and to decrease the risk of anaphylactic transfusion reactions in such patients [189]. There are also some cases that illustrate successful desensitization to IgA using IgA-enriched immunoglobulin preparations as a source of antigen [190].

### 6.3. Malignancy

There is a relationship between the occurrence of immune disorders and the overall risk of malignancy. This relationship is evident in some immunity disorders such as CVID [191,192]. It has been shown that the risk of malignant lymphoma among these patients is increased by 30 times, while the risk of gastric cancer is 47 times higher [193]. In the case of IgA deficiency, the association with the incidence of cancer is not that clear. There are studies that report the different incidences of malignancies among patients with sIgAD, especially adenocarcinoma of the gastrointestinal tract, and lymphomas. Such studies require a long-term follow-up to diagnose this kind of evolution, therefore there are not much data regarding this topic. In a review of 330 patients, the authors report a 1.5% prevalence of malignancy, and five patients who presented neoplasms had Hodgkin lymphoma, acute lymphoid leukemia, Wilms tumor, Burkitt lymphoma, and ganglioneuroma [132]. Another study that included 63 children from Israeli reports a much higher frequency of malignancy (4.8%) [99] and the following malignancies were present: astrocytoma, adenocarcinoma of the colon, Hodgkin’s lymphoma, neuroblastoma. On the other hand, a combined Danish and Swedish study including 386 patients with sIgAD did show an elevated incidence of cancer compared to a healthy cohort, however, this increase was non-significant [194]. In 2015, Ludvigsson et al. performed a prospective nationwide population-based cohort study with 2320 individuals with IgA deficiency [195]. They concluded that there is a moderately increased risk of cancer, especially gastrointestinal one, and that the risk is highest after diagnosis of sIgAD.

## 7. Atopic Diseases in Other PIDs

Tuano et al. described the prevalence of asthma, allergic rhinitis, atopic dermatitis, and food allergy in a cohort of 2923 patients with PID in US population [123]. Atopic dermatitis and food allergy were most common in patients with CVID, combined immunodeficiency (CID), and hyper IgE syndrome. Patients with CID and sIgAD presented a higher percentage of food allergy symptoms than the healthy population; 33.3% in CID and 25% in SIgAD [123]. In CVID the prevalence rates of asthma, rhinitis, and documented food allergy have been established as 37.5%, 55.5%, and 11.25% respectively [196]. In the case of patients with hypogammaglobulinemia, prevalence rates of asthma, rhinitis, and atopic dermatitis were established at 20%, 22%, and 9% respectively. Interestingly, Szczawinska-Poplonyk assessed the incidence of food allergy as 74% in the pediatric population [131,197,198].

## 8. Conclusions

sIgAD is an antibody deficiency and it usually remains undiagnosed throughout the patient’s life, due to its frequent asymptomatic course. If symptomatic, sIgAD is connected to more frequent viral and bacterial infections of upper respiratory, urinary, and gastrointestinal tracts, as well as allergic and autoimmune diseases. It was suggested that allergic diseases (e.g., eczema, rhinitis, asthma) are more common in patients diagnosed with this particular PID, however, the prevalence and severity of allergic manifestations can be associated with ethnic background.

Since there is a controversy in the scientific world whether the coexistence of IgA deficiency and allergic diseases is in fact true, further studies on a large group should be carried out. Atopy in sIgAD subjects is common, but is also possible that it is overlooked. Therefore, it is necessary to follow diagnostic criteria to make a diagnosis of any atopic disease. The possible reasons for different prevalence which is observed in different studies could be caused by different diagnostic criteria or inclusion of patients based on parentally reported symptoms in the children population.

Moreover, these patients can present with simultaneous atopic and infectious manifestations which can intensify the symptoms; therefore, atopic diseases should be treated aggressively in patients with sIgAD. It is necessary to provide these patients with a proper multi-disciplinary team of physicians.

Moreover, it is important to emphasize that the course of the disease may change and there are some serious complications of this disorder, among which there are progression to CVID, transfusion-related anaphylaxis, and malignancy. Although they do not happen very often, sIgAD remains the most common PID disease, therefore physicians should be aware of all possible complications to provide the best care for their patients.

## Figures and Tables

**Figure 1 jcm-10-03809-f001:**
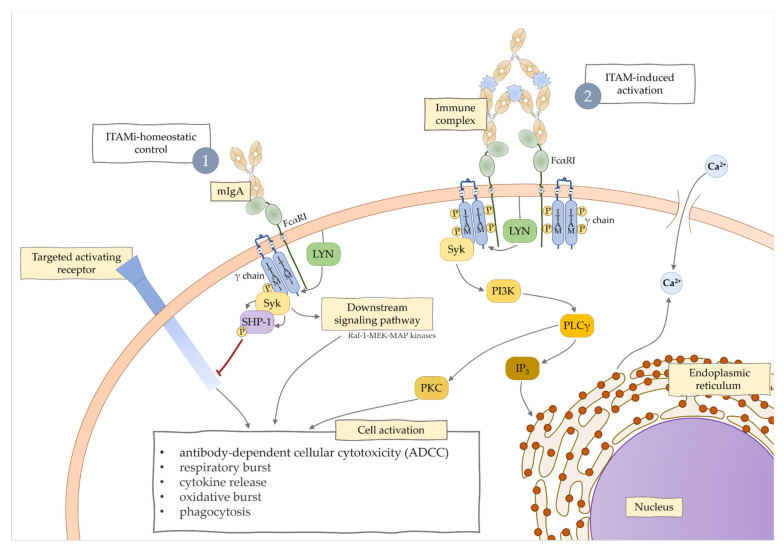
Regulation of immune responses by FcαRI, including ITAM-induced activation and ITAMi-control.

**Table 4 jcm-10-03809-t004:** Allergic rhinitis/conjunctivitis and sIgAD.

Year	Country	Sample Size	Disease Prevalence Among SIgAD (%)	Reference	Diagnostic Tools
2008	Brazil	126	53.2 (AR)	[79]	Diagnostic criteria of allergic diseases were not defined in the paper
2009	Iran	37	40 (AR/C)	[10]	The allergy status was evaluated by skin prick test, using 14 common standard allergen extracts
2010	Israel	63	12.7 (AR)	[99]	Retrospective study of patients records
2012	Spain	330	9 (AR)	[132]	Retrospective study of patients records
2013	Iceland	32	37.5 (AR/C)	[137]	Self-administered questionnaire + interview performed by physician + skin prick tests
2017	Turkey	81	27.2 (AR)	[101]	Presence of allergic rhinitis was only accepted if it was diagnosed by a physician
2019	Italy	103	18.4 (AR) 9.7 (C)	[142]	Patients’ clinical data were collected at enrolment and every 6–12 months for 5 years. Diagnostic criteria for allergic diseases were not defined in the paper

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
