# Peer review of "The Epidemiology and Clinical Presentations of Atopic Diseases in Selective IgA Deficiency"

_jcm, 2021, doi:10.3390/jcm10173809_

Round 1
Reviewer 1 Report
This is a comprehensive review of the epidemiology and clinical presentations of atopic disease in patients with Selective IgA deficiency. It discusses the breadth of allergic disease commonly seen in SIgAD - notably atopic dermatitis, asthma, food allergy, allergic rhinitis and anaphylaxis to blood products.
- Generally speaking I think this review has too much background/review information about PIDs and could focus the introduction better. Aside from the first paragraph which introduces the concept of PID, there is no need to detail combined immunodeficiency, complement, phagocytic defects and rather this should only review humoral defects since IgA deficiency in that type.
- The IGA structure section could be condensed.
- Page 3, line 148 lists 5 phenotypes of SIgAD. This should be the same order as they are described in the text but is not. Allergy is listed 2nd but described 4th in text.
- Section 6.2 should also discuss the other specific antibodies that are thought to play a role in anaphylactic reactions (e.g. anti-IgA IgE)
Author Response
Dear Reviewer,
Thank you for devoting you time and effort to review our paper on “The Epidemiology and Clinical Presentations of Atopic diseases in Selective IgA Deficiency”. We have corrected our paper according to yours suggestions and below we present point-by-point answers to the concerns.
Reviewer 1
This is a comprehensive review of the epidemiology and clinical presentations of atopic disease in patients with Selective IgA deficiency. It discusses the breadth of allergic disease commonly seen in SIgAD - notably atopic dermatitis, asthma, food allergy, allergic rhinitis and anaphylaxis to blood products.
Comment: “Generally speaking I think this review has too much background/review information about PIDs and could focus the introduction better. Aside from the first paragraph which introduces the concept of PID, there is no need to detail combined immunodeficiency, complement, phagocytic defects and rather this should only review humoral defects since IgA deficiency in that type.”
Response: We tried to condense the background information by removing some sentences that were not pivotal for the subject of the review.
Comment: “The IGA structure section could be condensed.”
Response: We have removed several phrases in this section, according to the Reviewer suggestion:
“IgA is a class of immunoglobulins characterized by the presence of an alpha heavy chain. The daily synthesis of immunoglobulin A exceeds the total production of all other immunoglobulins [9]. In the human body, there are two subclasses of this immunoglobulin: IgA1 and IgA2. The most important difference between them lays in the structure of their hinge region and the number of the glycosylation sites [10]. IgA1 has an elongated hinge region and, comparing to IgA2, has more sialic acid, whereas IgA2's hinge region is shortened to just 7 amino acids [11,12]. These properties lead to the difference in their effect on immune cells.
In serum, IgA1 is predominant, accounting for as much as 90%, while in mucosal tissues, both subclasses are more evenly distributed, comprising 40% IgA1 and 60% IgA2. [13]. In human blood, IgA occurs mostly in monomeric form, while secretory IgA (SIgA) present on the surface of mucous membranes usually occurs in the form of dimers, much less often as trimers and tetramers [14,15]. Dimeric SIgA antibodies, covalently linked by a J-linking chain, are secreted onto the mucosal surface with their characteristic secretory complement (SC). Both the SC and polymeric immunoglobulin receptor (pIgR) are involved in transcytosis, a specific process of IgA transport from the basal and lateral surfaces of epithelial cells [15].”
Comment: “Page 3, line 148 lists 5 phenotypes of SIgAD. This should be the same order as they are described in the text but is not. Allergy is listed 2nd but described 4th in text.”
Response: Thank you for this suggestion. We have changed the order: “Yazdani et al. [6] in their work from 2015 divided these phenotypes into five main categories: asymptomatic, minor infections, autoimmunity, allergy, severe.”
Comment: “Section 6.2 should also discuss the other specific antibodies that are thought to play a role in anaphylactic reactions (e.g. anti-IgA IgE)”
Response: We have added information on anti-IgA IgE as suggested by the Reviewer (https://pubmed.ncbi.nlm.nih.gov/21835445/, https://jcp.bmj.com/content/54/5/337)
Once again thank you for your time and consideration.
Kind regards,
Paulina Niedźwiedzka-Rystwej
Reviewer 2 Report
This is a well-written review on an interesting topic. For specific points see line-by-line review below, however, in general, minor improvements to the English language may be welcome (not being a native speaker I don't feel fully qualified to judge this, but some misspelled words or weird formulations were apparent). I would also request the authors to employ higher stringency and scientific rigor when analysing data of other studies - for the meta-analysis which this review attempts, there was no discussion of methodology of the other studies, which may be extremely important in driving the large differences in published prevalences of atopic disease in sIgAd. Due to my background I would also welcome deeper discussion of the relationship between sIgAdef and other antibody deficiences, especially from a clinical/diagnostic standpoint.
Also, I would suggest consideration of the authorship collective - for a review-type article with no original data to have 14 authors seems excessive.
Additionally, the author contributions list RH for visualisation, but there are no figures in the manuscript?
In general however I was mostly satisfied with this manuscript.
31 egzema
39 models of inheritance
40 polygenic rather than multi-genetic, which is not widely used
42 I would suggest mentioning cancers as a complication of PIDs
44 according to the IUIS classification from 2020, PIDs are classified into 10 different groups. Suggesting division into 4 groups is simplistic and at this day obsolete. please formulate differently.
50 unfortunately only some aspects of primary antibody disorders can be treated through IG replacement therapy. please mention or re-word.
58 there are many other reasons for delay in diagnosis, including low access to sequencing facilities, long pipelines, variants of unknown significance, confounding effect of childhood infections etc. use more careful wording.
102 this is interesting and I think it would be helpful to elaborate slightly about hwhich cells express FcaRI and how is the suppression mediated.
111, 113 complement, not component
111 and further, it would be helpful to elaborate on the mechanisms how IgA suppresses IL-6, promoted IL-10, blocks classical complement pathway, and especially how it can affect oxidative burst and phagocytosis in the mucosa.
126 what kind of mRNA, do you mean low number of IgA transcripts?
127 these cytokines are involved in Th1 and Th2 responses as well, why would they affect IgA production in particular, but not the other pathways?
133 elaborate on the proposed T cell side of sIgAd please. One study on Tregs is insufficient, the relationship between Tregs and IgA would need more discussion.
139 good and imporant paragraph, thank you
153 how many develop clinical manifestations, over how long period, and what manifestations? too vague.
172 what degree of IgM increase? significant? is it during steady state, or perhaps ongoing immune stimulation with mucosal pathogens may drive low level inflammation and IgM production?
174 what about norovirus and enterovirus, which are commonly seen in other antibody deficiencies, CVID and Bruton, respectively?
184 what about immunomodulation, bacterial lysates, beta-glucans, isoprinosine etc? any evidence on this?
197 correlation between prevalence of sIgAdef and COVID-19 in general population is perhaps very interesting, but very weak evidence. any studies on IgA in COVID-19 admitted patients compared to general population?
224 I would mention that "actude diseases could induce autoimmunity IN sIgAD, which was not clearly apparent to me from this sentence.
226+ comparing different studies of allergic manifestations, it's absolutely crucial to discuss their methodology as well. how was allergy diagnosed? self-reporting, sIgE, skin testing, in case of alelrgy, was it elimination/exposition tests, double blind placebo controlled food challenges? how were the cohorts recruited, were those patients who were first examined due to allergy, where sIgAd was found, or were they first examined due to infections and then developed allergy? this is absolutely vital for the data discussed further in the review and particularly important for outlier studies, such as the Iranian study. mention these in the tables (at least how was the food allergy diagnosed).
257 ITAMs abbreviation not used previously, unless I'm mistaken?
259 TGF-beta
266 citation needed
269 what kind of immunotherapy, and administered why? also citation needed.
283 few = how many?
353, 370+ it would be helpful to add odds ratio for sIgAd vs general population in the respective studies. this would apply to all tables.
384 more discussion on the overlap between sIgAd and CVID would be welcome, given that they can progress from one to the other, and CVID diagnosis may also rely on low serum IgA (+IgG) levels. see also comment for line 153. differential diagnosis between severe sIgAd and other PIDs may be helpful to propose. I would also welcome comparison of atopic complications (main topic of this study) in sIgAdef and in other PIDs (such as the subclass deficiency, CVID or specific antibody deficiency).
412 weird citation for CVID diagnostic criteria? according to ESID and the ICON it's low IgG + low IgA/IgM + impaired vaccination response to protein or polysaccharide antigens.
415 please elaborate on monogenic CVID causes beginning as sIgAdef
420 how does B cell phenotype differ between CVID and sIgAdef? can this be used to differentiate patients who will progress to CVID?
450 more recent studies on CVID associated malignancies exist, please cite one of those.
475 egzema again
Author Response
Dear Reviewer,
Thank you for devoting you time and effort to review our paper on “The Epidemiology and Clinical Presentations of Atopic diseases in Selective IgA Deficiency”. We have corrected our paper according to your suggestions and below we present point-by-point answers to the concerns.
Reviewer 2
This is a well-written review on an interesting topic. For specific points see line-by-line review below, however, in general, minor improvements to the English language may be welcome (not being a native speaker I don't feel fully qualified to judge this, but some misspelled words or weird formulations were apparent). I would also request the authors to employ higher stringency and scientific rigor when analysing data of other studies - for the meta-analysis which this review attempts, there was no discussion of methodology of the other studies, which may be extremely important in driving the large differences in published prevalences of atopic disease in sIgAd. Due to my background I would also welcome deeper discussion of the relationship between sIgAdef and other antibody deficiences, especially from a clinical/diagnostic standpoint.
Also, I would suggest consideration of the authorship collective - for a review-type article with no original data to have 14 authors seems excessive.
Additionally, the author contributions list RH for visualisation, but there are no figures in the manuscript?
Response: We would like to mention that among the 12 authors of this manuscript, there are beginning scientists who contributed to the work, however, the paper required in-depth analysis and improvement by more experienced people. As such, the number of authors mentioned in our manuscript might seem high, but all of them deserved to be included. Also, we have included a figure in the current form of the manuscript, therefore contribution for visualisation is justified.
In general however I was mostly satisfied with this manuscript.
Comment: “31 egzema, 475 egzema again”
Response: We have changed the spelling from egzema to eczema, thank you for noticing.
Comment: “39 models of inheritance”
Response: We agree with the Reviewer. We have corrected that.
Comment: “40 polygenic rather than multi-genetic, which is not widely used”
Response: We agree with the Reviewer. We have corrected that.
Comment: “42 I would suggest mentioning cancers as a complication of PIDs”
Response: We agree with the Reviewer. We have mentioned specifically cancers as a complication of PIDs: “ They weaken the body's defenses, increasing the frequency of infections as well as the risk of autoimmune and proliferative diseases, including cancers. “
Comment: “44 according to the IUIS classification from 2020, PIDs are classified into 10 different groups. Suggesting division into 4 groups is simplistic and at this day obsolete. please formulate differently.”
Response: We agree with the Reviewer. We reformulated this.
Comment: “50 unfortunately only some aspects of primary antibody disorders can be treated through IG replacement therapy. please mention or re-word.”
Response: We agree with the Reviewer. We have deleted that part.
Comment: “58 there are many other reasons for delay in diagnosis, including low access to sequencing facilities, long pipelines, variants of unknown significance, confounding effect of childhood infections etc. use more careful wording.”
Response: We agree with the Reviewer. We have deleted that part.
Comment: “102 this is interesting and I think it would be helpful to elaborate slightly about hwhich cells express FcaRI and how is the suppression mediated.”
Response: We agree with the Reviewer. We added more detailed explanation:
Comment: “111, 113 complement, not component”
Response: Thank you for noticing this mistake. We corrected that.
Comment: “111 and further, it would be helpful to elaborate on the mechanisms how IgA suppresses IL-6, promoted IL-10, blocks classical complement pathway, and especially how it can affect oxidative burst and phagocytosis in the mucosa.”
Response: We have added more specific information about mentioned mechanisms.
Comment: “126 what kind of mRNA, do you mean low number of IgA transcripts? “
Response: Both secreted and membrane Ca mRNA in IgA-switched B cells
Comment: "127 these cytokines are involved in Th1 and Th2 responses as well, why would they affect IgA production in particular, but not the other pathways?"
Response: We have added scientific data about the most important cytokines involved in IgA production in patients with SIgAD.
Comment: “133 elaborate on the proposed T cell side of sIgAd please. One study on Tregs is insufficient, the relationship between Tregs and IgA would need more discussion.”
Response: We have added additional information about Tregs and SIgAD.
Comment: “139 good and imporant paragraph, thank you”
Response: Thank you very much.
Comment: “153 how many develop clinical manifestations, over how long period, and what manifestations? too vague.”
Response: Unfortunately, we found it difficult to find some specific information about the follow-up of asymptomatic patients with sIgAD. However, we have added results from an Italian study, where authors present the long period evaluation of symptoms of their patients, which gives some evidence, that the course of the disease is changing with time.
Comment: “172 what degree of IgM increase? significant? is it during steady state, or perhaps ongoing immune stimulation with mucosal pathogens may drive low level inflammation and IgM production?”
Response: We have added more specific data.
Comment: “174 what about norovirus and enterovirus, which are commonly seen in other antibody deficiencies, CVID and Bruton, respectively?”
Response: Various GI tract infections with viruses and bacteria are commonly seen in sIgAD. We have reformulate the phrases, to avoid false impression, that the acute diarrhoea in patients with sIgAD is caused only by G.lamblia.
“Besides, patients with sIgAD suffer from urinary tract infections (UTI) and gastrointestinal tract infections with viruses and bacteria. Moreover, intermittent or chronic diarrhoea due to Giardia Lamblia is common, because the attachment and proliferation of this parasite on the gastrointestinal mucous are facilitated due to lack of IgA [56,57]”
Comment: 184 what about immunomodulation, bacterial lysates, beta-glucans, isoprinosine etc? any evidence on this?
Response: Thank you, we have added some information about this interesting topic.
Comment: 197 correlation between prevalence of sIgAdef and COVID-19 in general population is perhaps very interesting, but very weak evidence. any studies on IgA in COVID-19 admitted patients compared to general population?
Response: We have added some information about the topic, however we did not found any multicentered studies about IgA deficiency in COVID-19 admitted patients, only one based on the Israeli population.
Comment: 224 I would mention that "actude diseases could induce autoimmunity IN sIgAD, which was not clearly apparent to me from this sentence.
Response: Indeed, we have changed this unfortunate sentence.
Comment: “226+ comparing different studies of allergic manifestations, it's absolutely crucial to discuss their methodology as well. how was allergy diagnosed? self-reporting, sIgE, skin testing, in case of alelrgy, was it elimination/exposition tests, double blind placebo controlled food challenges? how were the cohorts recruited, were those patients who were first examined due to allergy, where sIgAd was found, or were they first examined due to infections and then developed allergy? this is absolutely vital for the data discussed further in the review and particularly important for outlier studies, such as the Iranian study. mention these in the tables (at least how was the food allergy diagnosed).”
Response: We agree with the Reviewer. We have added the data of methodology in the tables.
Comment: “257 ITAMs abbreviation not used previously, unless I'm mistaken?”
Response: We agree with the Reviewer. In the original manuscript ITAM had not been used. However, in the current version it was mentioned for the first time in the “IgA - function” section, and we included the meaning of the abbreviation.
Comment: “259 TGF-beta”
Response: We have corrected that, sorry for the mistake.
Comment 266, 269
Response: We have decided to describe other preventive form of SIgAD treatment in line 184.
Comment: “283 few = how many?”
Response: four
“but there were only four patients with sIgAD in the registry.”
353, 370+ it would be helpful to add odds ratio for sIgAd vs general population in the respective studies. this would apply to all tables.
384 more discussion on the overlap between sIgAd and CVID would be welcome, given that they can progress from one to the other, and CVID diagnosis may also rely on low serum IgA (+IgG) levels. see also comment for line 153. differential diagnosis between severe sIgAd and other PIDs may be helpful to propose. I would also welcome comparison of atopic complications (main topic of this study) in sIgAdef and in other PIDs (such as the subclass deficiency, CVID or specific antibody deficiency).
Comment: “412 weird citation for CVID diagnostic criteria? according to ESID and the ICON it's low IgG + low IgA/IgM + impaired vaccination response to protein or polysaccharide antigens.”
Response: Diagnostic criteria from ESID includes: marked decrease of IgG and marked decrease of IgA with or without low IgM levels and at least one of the following: poor antibody response to vaccines or switched memory B cells (<70% of age-related normal value). In our manuscript we included criteria from ESID:
“Common variable immunodeficiency (CVID) is an immune disorder characterized by decreased serum levels of both IgG and IgA, with or without a decreased level of IgM, and poor antibody vaccine response or low switched memory B cells less than 70% of age-appropriate normal [Seidel_2019].”
Comment: “415 please elaborate on monogenic CVID causes beginning as sIgAdef”
Response: We have added a paragraph about single gene mutations in CVID and we instanced mutation in TACI as the one which is present in CVID/sIgAD families.
420 how does B cell phenotype differ between CVID and sIgAdef? can this be used to differentiate patients who will progress to CVID?
Response: We have added information about B-cell phenotypes. However, in our opinion, this topic is not described well enough to answer the question as to whether it is possible to differentiate patients in danger of CVID progression based only on B-cell phenotyping.
450 more recent studies on CVID associated malignancies exist, please cite one of those.
Response: Cited
Once again, we would like to thank you for the time, effort, and consideration of our paper.
Kind regards,
Paulina Niedźwiedzka-Rystwej